# Novel Vaccine against Pathological Pyroglutamate-Modified Amyloid Beta for Prevention of Alzheimer’s Disease

**DOI:** 10.3390/ijms24129797

**Published:** 2023-06-06

**Authors:** Karen Zagorski, Olga King, Armine Hovakimyan, Irina Petrushina, Tatevik Antonyan, Gor Chailyan, Manush Ghazaryan, Krzysztof L. Hyrc, Jean Paul Chadarevian, Hayk Davtyan, Mathew Blurton-Jones, David H. Cribbs, Michael G. Agadjanyan, Anahit Ghochikyan

**Affiliations:** 1Department of Molecular Immunology, Institute for Molecular Medicine, Huntington Beach, CA 92647, USA; kzagorski@immed.org (K.Z.); osvystun@uci.edu (O.K.); ahov@immed.org (A.H.); tantonyan@immed.org (T.A.); gorchailyan@gmail.com (G.C.); manush@immed.org (M.G.); magadjanyan@immed.org (M.G.A.); 2Institute for Memory Impairments and Neurological Disorders, University of California, Irvine, Irvine, CA 92697, USA; ipetrush@uci.edu (I.P.); jchadare@uci.edu (J.P.C.); hdavtyan@uci.edu (H.D.); mblurton@uci.edu (M.B.-J.); cribbs@uci.edu (D.H.C.); 3The Hope Center of Neurological Disorders, Washington University School of Medicine, St Louis, MO 63110, USA; hyrck@wustl.edu; 4Department of Neurobiology & Behavior, University of California, Irvine, Irvine, CA 92697, USA

**Keywords:** Alzheimer’s disease, vaccine, post-translational modification, pyroglutamate, prevention

## Abstract

Post-translationally modified N-terminally truncated amyloid beta peptide with a cyclized form of glutamate at position 3 (pE_3_Aβ) is a highly pathogenic molecule with increased neurotoxicity and propensity for aggregation. In the brains of Alzheimer’s Disease (AD) cases, pE_3_Aβ represents a major constituent of the amyloid plaque. The data show that pE_3_Aβ formation is increased at early pre-symptomatic disease stages, while tau phosphorylation and aggregation mostly occur at later stages of the disease. This suggests that pE_3_Aβ accumulation may be an early event in the disease pathogenesis and can be prophylactically targeted to prevent the onset of AD. The vaccine (AV-1986R/A) was generated by chemically conjugating the pE_3_Aβ_3-11_ fragment to our universal immunogenic vaccine platform MultiTEP, then formulated in Advax^CpG^ adjuvant. AV-1986R/A showed high immunogenicity and selectivity, with endpoint titers in the range of 10^5^–10^6^ against pE_3_Aβ and 10^3^–10^4^ against the full-sized peptide in the 5XFAD AD mouse model. The vaccination showed efficient clearance of the pathology, including non-pyroglutamate-modified plaques, from the mice brains. AV-1986R/A is a novel promising candidate for the immunoprevention of AD. It is the first late preclinical candidate which selectively targets a pathology-specific form of amyloid with minimal immunoreactivity against the full-size peptide. Successful translation into clinic may offer a new avenue for the prevention of AD via vaccination of cognitively unimpaired individuals at risk of disease.

## 1. Introduction

Various immunotherapeutic approaches for treating and preventing Alzheimer’s Disease (AD) have been developing for nearly a quarter century, yet an effective vaccine remains elusive [1,2,3]. One of the key issues with anti-Aβ immunotherapies is the inadequately late start of the treatment. Since the aggregation of Aβ and even neuronal loss occur years before the first cognitive symptoms, a successful therapeutic approach targeting Aβ needs to be initiated long before the onset of the symptoms and well before the accumulation of the pathological tau protein [4,5,6]. The treatment of cognitively unimpaired people at risk of disease would need to be extremely safe, minimally invasive, and cost-effective to be viable. Yet, most immunotherapies currently in development are based on passive immunizations with high titers of exogenous humanized antibodies, which are costly, require regular infusions, and are often associated with serious side effects [7].

Vaccination, as opposed to passive immunotherapy, promotes the endogenous production of therapeutically potent antibodies for extended periods. Vaccines do not require monthly re-administration, which makes vaccination more affordable and less invasive [8]. These properties can address many of the limitations mentioned above and emphasize the need for a vaccine aimed at pathology-specific targets in cognitively unimpaired people at risk of AD.

One of the most promising targets is the N-terminally truncated pE_3_Aβ. The pE_3_Aβ species are present in great abundance in AD-associated plaques [9,10,11] and form soluble oligomers that potentially seed full-length Aβ_1–42_. The pyroglutamate-modified peptides accumulate a decade prior to the symptom onset, yet their relative abundance in the AD brain is approximately threefold higher compared to the untruncated forms, making them a superior disease-specific target [12,13,14,15,16,17,18,19]. Furthermore, they are an active component of the amyloid cascade pathogenesis rather than a harmless bystander and capable of promoting the conversion of Aβ_42_ into toxic oligomers in a prion-like manner [20,21,22,23,24,25,26,27,28,29,30]. The most valuable evidence in favor of targeting pE_3_Aβ is the recent clinical trial performance of donanemab, a humanized IgG1 monoclonal antibody generated from mouse mAb, mE8-IgG2a [31,32]. Due to the rapid clearance of Aβ plaques and slowing of cognitive decline, donanemab has received Breakthrough Therapy designation from the FDA. However, despite being humanized, 90% of individuals treated with donanemab developed detectable anti-donanemab antibodies after a single injection, as opposed to 0.6% for currently approved aducanumab [33,34]. Subsequent administrations of donanemab can further increase the titers of these anti-donanemab antibodies, which may lead to diminishing efficacy of the drug and increased adverse events over time [35].

The general feasibility of vaccination against pE_3_Aβ with high selectivity against untruncated forms of Aβ was first demonstrated in 2009 [36]. Since then, AFFITOPE AD03 (Affiris, Inc., San Francisco, CA, USA) has been the only anti-pE_3_Aβ vaccine evaluated in a clinical trial. This study was terminated in 2013 after phase Ia completion in 2011, and no reports or peer-reviewed publications regarding the immunogenicity of this vaccine and the structure of mimotope peptide attached to KLH are available to this day [37]. A preclinical stage vaccine for pE_3_Aβ (amyloid peptide fused with a tetanus T helper cell epitope, P2) with ~16-fold selectivity compared to the full-length peptide was reported by Li et al. [38]. Recently, AC Immune reported that antibodies generated in mice and non-human primates by an optimized version of the ACI-24 vaccine targeting Aβ1–15 bind also to pE_3_Aβ due to the broad epitope coverage of the Aβ peptide [39]. However, this binding is not specific to a pyroglutamate-modified epitope of Aβ, and the titers of antibodies binding to pE_3_Aβ are significantly lower than those binding to the full-length Aβ_42_. Additionally, the “TAPAS” vaccine, which targets a stabilized cyclic form of Aβ1–14, has been shown to generate antibodies recognizing pE_3_Aβ. Antibodies generated by the TAPAS vaccine, as well as the TAP01 humanized antibody, showed therapeutic benefits in 5XFAD and Tg4–42 mouse models of AD [40].

In this study, taking advantage of the immunogenic and universal MultiTEP platform technology, we have developed and tested a vaccine candidate against pE_3_Aβ in a stringent 5XFAD mouse model of AD [41,42,43,44,45,46].

The pyroglutamated aa 3–11 peptide was synthesized with C-terminal azide and chemically attached to the specially engineered version of MultiTEP carrier protein using copper-free click chemistry. The data below demonstrate that the generated conjugate vaccine is highly immunogenic and highly selective for pyroglutamate modification. More importantly, the vaccine shows promising therapeutic potential in the transgenic 5XFAD mouse model of AD, reducing both pE_3_Aβ and full-length Aβ in their brain, despite only minimal immune response towards the full-size peptide generated by this novel vaccine.

## 2. Results

### 2.1. Preparation of the MultiTEP-Based Carrier Protein for Bioconjugation

The MultiTEP carrier protein was modified via a thiol-specific chemical reaction to enable the chemical attachment of post-translationally modified peptides, such as pyroglutamated Aβ_3–11_. First, the pre-existing cysteines within the protein were mutated to serines to avoid the chemical modification of the epitopes containing them, while maximally preserving their structural properties. Then, the attachment sites were introduced by adding a linker containing three cysteines flanked on either side with four repeats of lysine-glutamate (KEKEKEKE). The highly hydrophilic linker was chosen to maximize the solvent exposure of the cysteines and reduce the steric hindrances during the chemical conjugations, allowing for up to three peptides to be attached to each molecule of MultiTEP. The carrier was then conjugated to the azide-labeled pEAβ_3–11_ peptides using a Maleimide-PEG4-DBCO heterobifunctional crosslinker reagent.

The data presented in Figure 1 demonstrates the Coomassie-stained SDS PAGE, along with a western blot visualized with commercial anti-pE_3_Aβ polyclonal antibodies. The unconjugated carrier can be seen as a single band on the Coomassie stain, but not on the western blot since it does not contain the target epitopes. The conjugation product, on the other hand, is more heterogeneous with three defined bands most likely corresponding to the protein carrying 1–3 copies of the target epitope. Higher molecular weight species were also observed and likely oligomers of the conjugated proteins stained by the commercial anti-pE_3_Aβ polyclonal antibody.

### 2.2. Choosing the Mouse Model of AD

Evaluation of the 5XFAD mouse model of AD demonstrated a significant increase of the pE_3_Aβ in the cortex and hippocampus area at 6 months that further increases with age [22,47,48,49,50]. Accordingly, we bred these familial AD transgenic mice overexpressing human *APP695* with Swedish, Florida, and London mutations, along with two mutations in the human PS1 in our vivarium and verified the presence of pE_3_Aβ depositions at eight months of age. Brain slices from the 5XFAD mouse model were imaged using confocal microscopy with fluorescently labeled antibodies specific to full-length Aβ_42_ or N-terminally truncated pE_3_Aβ. Signals from the two antibodies were collected in separate color channels and the images were merged as two pseudocolors to identify the relative localization and quantities of each form of Aβ. Figure 2A demonstrates that both forms of the amyloid protein were present in the brains of the animals, with the pyroglutamated form located primarily within the plaque core rather than in the diffuse regions seen in human pathology [51]. Since the 5XFAD animals bred in our animal facility had detectable deposits of pE_3_Aβ, they have been deemed a valid model for testing vaccination-mediated clearance of this pathology.

### 2.3. Immunogenicity and Selectivity of the AV-1986R/A Vaccine in 5XFAD Mice

The 5XFAD mice were sorted into 3 groups, 12 males and 12 females in each, with 2 control groups (PBS or Advax^CpG^ adjuvant) and 1 experimental group for the vaccine candidate termed AV-1986R/A, consisting of the conjugation product formulated in the Advax^CpG^ adjuvant. The detailed schedule of immunizations and study can be seen in Figure 2B. The sera collected after the fourth immunization were evaluated by ELISA to quantify the antibody titers against the pE_3_Aβ peptide, as well as against the unmodified Aβ42. The endpoint titers against the pyroglutamate-modified peptide were significantly higher (*p* < 0.0001 Mann-Whitney U test), with the geometric mean titers being 74-fold higher against the pE_3_Aβ. The effect was similar for male and female animals, with a slightly higher, but statistically insignificant, geometric mean ratio of the endpoint titers in females. The data is plotted in Figure 3.

To further evaluate the pE_3_Aβ selectivity of the generated antibodies, we used an overlapping peptide library in competition ELISA. The data presented in Appendix A demonstrates that none of the alternative peptides efficiently bind to the pyroglutamate-specific antibodies, including the truncated Aβ_3–14_ lacking the cyclic pE residue. However, the minor Aβ_42_-specific antibodies were efficiently inhibited with peptides containing the residues 3–11, with the epitope likely spanning residues 3–11 or 4–11. The antibody titers after each immunization were plotted to demonstrate the immune response kinetics and can be found in Appendix A. Finally, isotyping was performed for the major immunoglobulin types shown in Appendix A.

### 2.4. Immunohistochemical Evaluation of Plaque-Clearing in 5XFAD Mice

Four weeks after the fifth immunization, mice were euthanized at the final age of 6.5 months old. The brains were harvested from all three groups for further assessment of vaccine efficacy, with one-half of each brain being sliced for immunohistochemical characterization. The microslices were first evaluated for pyroglutamated amyloid (anti-pE_3_Aβ positive) to confirm primary target engagement and clearance. The positive area was compared between the three groups, and the data is plotted in Figure 4 along with representative images from AV-1986R/A-vaccinated- and control-animals. For both sexes, the tissue slices from the animals vaccinated with AV-1986R/A showed significantly less pE_3_Aβ-specific staining when compared to either control (*p* < 0.0001 U test). This indicates that the generated antibodies can successfully remove the targeted pathological species within the brain with high efficacy.

Next, tissue slices were visualized with antibodies specific to non-pyroglutamated Aβ, specifically 6E10 mAb, which we previously used to detect the non-pyroglutamated plaques in the 5XFAD animals. Again, the tissue from the vaccinated animals showed a significant reduction of the positive area compared to either control, regardless of sex, as shown in Figure 5.

### 2.5. Aβ Content of 5XFAD Mouse Brain Extracts by ECLIA

Half of each brain was extracted to obtain soluble and insoluble fractions of Aβ for their quantitative analysis via the MesoScale Discovery electrochemiluminescent immunoassay (ECLIA) kit. The data obtained from the analysis of the soluble fractions are shown in Figure 6. The only statistically significant differences observed were between the male Advax^CpG^-injected and AV-1986R/A-vaccinated animals, with a slight elevation of the soluble Aβ in the vaccine group, potentially suggesting solubilization of the plaque material by the antibodies. This effect may have been enhanced by the complex effects of the TLR9 agonists on brain homeostasis, which have been observed previously in the context of amyloid and AD [52,53,54,55,56].

The remaining insoluble fractions of the brain extracts were further solubilized with formic acid and assayed using the same method. The data obtained from the analysis of the insoluble fractions are plotted in Figure 7. Vaccinated animals showed a significant reduction of insoluble amyloid when compared to either control. Interestingly, the reduction was only significant in females. Despite the absence of statistical significance by the U test, the amyloid load in half of the vaccinated males (but none from controls) was below the minimum limit of quantitation of the assay, suggesting that the absence of statistical significance was due to the insufficient statistical power of the experimental design. This is, in general, a limitation that can be seen with various mouse models of AD that are only partially replicating the very complex human Alzheimer’s disease. 

## 3. Discussion

Alzheimer’s disease is one of the most challenging public health emergencies of our time. Currently, an estimated 35 million people live with AD globally [57]. Since age is the primary risk factor for developing AD, the increasing healthcare quality and life expectancy inevitably leads to an increased AD morbidity. Affected individuals undergo gradual cognitive deterioration and eventually require assistance for basic tasks, such as eating and bathing. In the final stages, the degenerative changes in the brain can perturb their ability to swallow, cough, and breathe. Along with the obvious human pain and suffering, AD also has an immense economic footprint for developed countries: just in the United States, where about 6.7 million people live with AD, the costs are estimated to be around $600 billion, which is why the research of treatments for AD is of strategic importance [58]. Since Aβ aggregation is currently considered the primary driving factor of AD pathogenesis, a significant portion of research efforts are aimed at restoring and protecting the normal homeostasis of Aβ in the brain. However, to-date, no reliable means of prevention or treatment exists for this debilitating disorder.

Recently, some success has been made in the realm of AD immunotherapy. More specifically, two monoclonal antibodies targeting Aβ oligomers (fully human, aducanumab) and protofibrils (humanized, lecanemab) were approved by the FDA. A third antibody targeting pyroglutamate-modified pE_3_Aβ (donanemab) is expected to also be approved. Despite these positive results, the overall epidemiological situation is still dire. It is important to understand that while the efficacy of the monoclonals is quantifiable, it is hardly meaningful in terms of an added quality of life. 

Our group has long been a proponent of a more proactive approach to AD immunotherapy, using active vaccines to prevent the onset rather than ameliorate or treat the symptoms. We have demonstrated, in animal models, that by the time the disease is symptomatic, the neuronal damage and tau-pathology are self-sustaining, and removal of the amyloid has little to no effect. In this paradigm, the applicability of monoclonal antibodies becomes questionable since they require frequent and costly injections that would not be appealing or affordable to the general population as a preventative measure. 

Vaccines are far more viable as a preventative therapeutic and have been used for exactly that purpose for centuries. Yet, no immunogenic, safe, and effective vaccine has yet been approved for AD. Multiple therapeutic vaccines against AD have failed in clinical trials due to poor immunogenicity, inefficacy, and serious adverse events [59]. These failures are part of a bigger trend in the field of anti-Aβ therapeutics, with antibodies and small molecules aimed at reducing production and increasing the elimination of Aβ peptides in people with prodromal (Mild Cognitive Impairment) or mild-moderate AD. Most of these therapeutics either did not ameliorate AD or worsened it [59,60,61,62,63,64]. We believe that one of the fundamental reasons for the failure of many anti-amyloid therapeutics stems from the obsolete notion that Aβ peptides are cellular waste with no physiological value or function, leading to approaches designed to indiscriminately target all isoforms and species of Aβ [65]. Current evidence suggests that Aβ peptides have diverse functions in normal brain homeostasis, and the complete depletion of these peptides from the brain affects neuroplasticity, memory, axonal regeneration, and other normal neuronal processes [66,67].

In this work, we chose to target a more pathology-specific isoform of Aβ to generate a safer vaccine for prophylactic use in healthy individuals. Specifically, we chose the post-translationally modified pE_3_Aβ based on the high pathogenicity and prevalence of this species in AD [9,11,13,15,17,18]. We hypothesize that targeting this species may lead to superior safety and efficacy, unlike previous Aβ vaccine efforts, by reducing the disease-specific Aβ.

To achieve robust and reliable immunogenicity, we applied our well-characterized universal vaccine platform MultiTEP, which is currently being evaluated in multiple preclinical IND enabling studies [42,43,44,46,68] and is in a first-in-human DNA vaccine Phase I clinical trial (NCT05642429). This vaccine platform is designed to induce strong immune responses in genetically diverse populations and potentially in immunosenescence individuals by activating pre-existing memory T helper cells with select universal foreign T helper (Th) epitopes. Such an approach is highly advantageous since the target population of pre-AD vaccines is generally 50+ years old who may have a senescent naïve Th cell pool, yet robust memory Th cells acquired in youth [69]. Additionally, using short, well-characterized universal epitopes allows for the minimalistic design of the MultiTEP platform, reducing potential safety concerns. Finally, the MultiTEP protein spontaneously assembles into oligomeric nanoparticles, thus protecting the target epitopes from rapid degradation, efficiently delivering them to antigen-presenting cells, and providing additional stimulation to the B cells due to the simultaneous presentation of multiple copies of the antigen [44]. Thus, our preclinical data demonstrated exceptionally potent and robust immune responses towards various neurodegeneration-associated target molecules [42,43,46], and, in this work, we presented the first MultiTEP-based conjugate vaccine for a post-translationally modified target—specifically, pE_3_Aβ.

The studies outlined in this manuscript demonstrate that our vaccine candidate induced robust and selective immune responses against the target isoform and led to a marked reduction of AD-like pathology in the 5XFAD mouse model of aggressive AD. We observed a more significant reduction of insoluble Aβ in female mice than in males. On the contrary, in our previous study, we observed a more substantial reduction of insoluble Aβ in the brains of male bigenic 5XFADxPS19 mice than in females when immunized with the Aβ vaccine formulated with the same adjuvant [46]. These gender-specific differences in vaccine efficacy in mouse models may arise from various factors, including environmental, genetic, and hormonal factors, and the severity of amyloid pathology.

It is important to note that the vaccine reduced not only pyroglutamated-Aβ plaques, but also non-pyroglutamated aggregates. This is especially promising since pE_3_Aβ is not an early component in 5XFAD mice as opposed to real AD pathogenesis in humans. This implies that the early clearance of pE_3_Aβ could be even more effective in ameliorating the downstream pathological events of the disease and shows particularly encouraging potential considering the current lack of preventive vaccinations for people at risk of AD. The next steps in developing this vaccine require further optimizing the chemistry for the conjugation of the pE_3_Aβ_3–11_ peptide to MultiTEP to improve yield and stability, as well as manufacturing of cGMP vaccine for preclinical safety-toxicology studies. Once completed, this data will allow us to apply for an IND and initiate a Phase 1 clinical trial for the assessment of the safety and immunogenicity of this novel drug product in humans.

## 4. Materials and Methods

### 4.1. Preparation of the MultiTEP-Based Vaccine for pE_3_Aβ by Click Chemistry

Prior reports on the structures of oligomeric and fibrillar species of amyloid showed that the region between residues 1 and 14 are highly surface-exposed and the tyrosine 10 is solvent exposed in Aβ_42_ oligomers to a similar extent to that found in the unfolded monomer [70,71]. These data and the epitope prediction tool described in [72] were used to determine and select the target peptide length [70,71,72].

A version of the MultiTEP protein with C-to-S substitutions within the T helper epitopes, as well as an N-terminal zwitterionic linker bearing three cysteines, was expressed and purified via chromatography on Nickel (II) nitrilotriacetate resin (Ni-NTA, Qiagen, Redwood City, CA), followed by a second purification on HisPur Cobalt superflow resin (ThermoFisher Scientific, Chino, CA). All purification steps were performed in the presence of 8M urea, and elution was performed by a gradual reduction of the pH. Fractions eluted at different pH points were collected, and SDS-PAGE, followed by Coomassie staining, was used to identify the fractions with the highest purity of the protein. SDS-PAGE was performed as described above, and Coomassie staining was performed as described previously [73]. The overall structure of the purified protein is shown in Appendix A.

Selected fractions were combined and transferred to 6 M guanidinium hydrochloride in PBS at pH 6.6. This slightly acidic condition ensures maximal selectivity of the maleimide coupling to thiols since the amines are only reactive in the deprotonated state. The cysteines were reduced by the addition of TCEP (tris(2-carboxyethyl)phosphine) at a molar ratio of 1:9 MultiTEP to TCEP, corresponding to a 3-fold molar excess of TCEP to cysteine residues [74]. The mixture was incubated for 2 h at 37 °C based on the reported reduction kinetics. Next, the DBCO-PEG4-Maleimide (Click Chemistry Tools, Scottsdale, AZ) linker was added at a 1:15 molar ratio of thiol to maleimide and allowed to incubate for 16 h at room temperature [72]. The product was purified by dialysis to remove the excess TCEP and DBCO-PEG4-Maleimide to prevent their reaction with the azide-labeled peptide. The azide-labeled synthetic peptide with the sequence pEFRHDSGYE-GGGGS-Azidolysine (Genscript, Piscataway, NJ) was added in a 1:6 molar ratio of MultiTEP to peptide and incubated for 24 h at 4 °C. The final purification was done by dialysis in 6 M guanidinium to remove the excess peptide, followed by refolding via gradual reduction of the guanidinium concentration in the dialysis buffer. Finally, the pH was adjusted to physiological in PBS, and doses were aliquoted and frozen. The summary of reactions is presented in Appendix A.

### 4.2. Mouse Model Selection

The 5XFAD mice (B6SJL-Tg(APPSwFlLon, PSEN1*M146L*L286V)6799Vas/-Mmjax) from Jackson Lab (Bar Harbor, ME, USA) were used in this study. These animals are maintained on a congenic C57Bl6/J background and co-expresses the human amyloid precursor protein (APP695) carrying the Swedish, Florida, and London mutations, as well as a human presenilin-1 (PS1) transgene carrying the M146L and L286V mutations under the Thy-1 promoter. Both APP and PS1 transgenes are co-integrated and thus, co-inherited. This transgenic mouse model was selected based on an aggressive pathology phenotype and rapid disease progression. These mice develop amyloid plaque as early as 2 months of age, which allowed for a faster experimental timeline. Most importantly, these animals develop deposits of pyroglutamated amyloid at 6 months of age [22,47,48,49].

Both female and male animals were used in this study. All animals were housed in a temperature and light-cycle-controlled facility. Their care was under the guidelines of the National Institutes of Health and an approved IACUC protocol at the University of California, Irvine.

### 4.3. Experimental Protocols

5XFAD mice were immunized with AV-1986R/A (20 μg/mouse/injection), formulated with an Advax^CpG^ adjuvant (Vaxine Pty Ltd., Adelaide, Australia) at 1 mg/mouse/injection. 

Control groups of 5XFAD mice were injected with an Advax^CpG^ adjuvant only or PBS. All mice were injected five times intramuscularly (Figure 2 B). Sera were collected 14 days after the fourth immunization, and anti-Aβ_42_ and anti-pE_3_Aβ antibody responses were analyzed. At the age of 6.5 months old, mice were terminated, and brains were collected for biochemical and immunohistological analysis.

### 4.4. ELISA

There were 96-well high binding plates separately coated with 1 μg/well (in 100 μL; Carbonate-Bicarbonate buffer, pH 9.6, o/*n* at 4 °C) of a pE_3_Aβ peptide (Genscript, Piscataway, NJ, USA) and unmodified Aβ_42_ (Genscript, Piscataway, NJ, USA). The next day, coated plates were washed 3 times and blocked with a blocking buffer (3% dry, non-fat milk in TBST, 200 μL/well, o/*n* at 4 °C). Sera samples were diluted 5-fold, serially starting at 1:1000 and 1:200 for the detection of anti-pE_3_Aβ and anti-Aβ_42_, respectively (in 0.3% dry, non-fat milk in TBST, 100 μL/well), then added to the plates and incubated o/*n* at 4 °C. The plates were washed 3 times with TBST, then the secondary antibody, HRP-conjugated goat anti-mouse IgG (Jackson ImmunoResearch Laboratories 1:2500 dilution) (cat#115-036-003), was added and incubated for 1 h at room temperature.

HRP-conjugated anti-IgG1, IgG2ab, IgG2b, and IgM-specific antibodies (Bethyl Laboratories, Inc., Montgomery. TX) were used to characterize the isotype profiles of anti-pE_3_Aβ antibodies in the pooled sera at a dilution of 1:1000 (Appendix A). 

Plates were washed 3 times with TBST before adding a 3,3′,5,5′-tetramethylbenzidine (TMB) substrate solution (Cat# 34029, ThermoFisher Scientific, Chino, CA, USA). The reaction was stopped after 5 min by adding 2N sulfuric acid. The OD at 450 nm was measured with a FilterMax F5 microplate reader. Endpoint titers of antibodies in mice sera were calculated as the reciprocals of the highest sera dilutions that gave an optical density reading thrice above the cutoff. The cutoff was determined as the titer of pre-immune sera at the same dilution.

### 4.5. Brain Extraction

Single hemispheres, previously frozen on dry ice and stored at −80 °C, were crushed on dry ice using a mortar and pestle. First, soluble brain fraction was extracted. Approximately 1/3 of each powdered hemisphere was homogenized in 250 µL of the protein extraction buffer (mixture of T-PER (ThermoFisher Scientific, Chino, CA, cat# 78510), 1× phosphatase inhibitors, and 1× EDTA (ThermoFisher Scientific, Chino, CA, REF 78426), 1× protease inhibitors (ThermoFisher Scientific, Chino, CA, REF 78429)). Then homogenates were centrifuged at 16,000× *g* for 15 min at 4 °C and supernatants were collected. Then, 150 µL of the protein extraction buffer was added to the pellets, and they were homogenized, centrifuged, and supernatants collected again to extract the rest of the soluble fraction. Final pellets were stored at −80 °C for further extraction of the insoluble fraction. Two soluble fractions were mixed, and the total protein concentration in the mixture was detected using a BCA assay. Then, 100 µL of 1 mg/mL of each soluble fraction was prepared using the T-PER buffer as a diluent. Stocks and 100 µL of 1mg/mL samples were frozen at −80 °C for further use. Samples were used for quantitative biochemical analysis (by ECLIA) of human Aβ at 25 µg per well. 

Second, insoluble brain fraction was extracted from the pellets stored at −80 °C. Then, 250 µL of 70% Formic Acid (FA) was added to each pellet. Pellets were centrifuged at 16,000 g for 30 min at 4 °C, and supernatants (insoluble fractions) were collected. Insoluble fractions were normalized based on the protein concentrations of soluble fractions obtained on the BCA assay. Then, 50 µL aliquots of normalized samples were frozen at −80 °C for consequent use. 

Prior to analyses, a neutralization buffer (1 M Tris base, 0.5 M NaH_2_PO_4_, 0.05% NaN_3_) was added to the normalized insoluble fractions. The pH of the samples was adjusted to 7 using 5N NaOH. Then, 10,000-fold dilutions of the neutralized samples were used for biochemical analyses. 

Quantitative biochemical analysis of human Aβ was conducted using a commercially available electrochemiluminescent multiplex assay system: Meso Scale Discovery (MSD, Rockville, MD). A V-PLEX human Aβ peptide panel (6E10 capture antibody) kit was used for simultaneous measurement of Aβ_38_, Aβ_40_, and Aβ_42_ in both soluble and insoluble protein fractions. 

### 4.6. Immunohistochemistry

Prior to the brain collection, mice were anesthetized with an intraperitoneal injection of a 150 mg/kg Nembutal Sodium Solution (Akorn, Inc., Lake Forest, IL, USA), then perfused with 36 mL of 1 × PBS (pH 7.4) using a peristaltic pump (12 mL/min). Brains were extracted and half-brains postfixed in 4% PFA for 48 h, washed in PBS, immersed in 30% sucrose for 48 h, and sectioned coronally at 40 μm thickness using the microtome (Leica SN2010R). Sections were collected into 12-well plates to obtain the series of equally-spaced sections throughout each brain, stored in PBS + 0.05% sodium azide. 

Prior to IHC, 12 equally spaced tissue sections from each half-brain were washed in Tris buffer, mounted onto slides, air-dried at RT, incubated at 60 °C for 20 min for better attachment, and cooled down to RT. Epitope retrieval was performed using a 0.1 M citrate buffer (Fisher Scientific, Waltham, MA, USA, Erpedia, cat # AP-9003-500) diluted according to the manufacturer’s instructions, and pre-warmed to 90 °C. The batches of slides carefully placed in incubation racks were immersed in the citrate buffer bath for 20 min at 90 °C. After cooling down to RT, slides were placed in a multi-sample staining tray and washed in 3 changes of Tris buffer for 5 min each, then once in Tris A (Thermo Scientific, Waltham, MA, USA, cat # 28360) diluted according to manufacturer’s instruction for 5 min, and once in Tris B (1% BSA in Tris A) for 10 min. 

For the detection of pE_3_-positive plaques, each incubation chamber (GRACE Bio-Labs, Bend, OR, USA, cat.# 645402) was filled up with 230 µL of an Abeta-pE_3_ antibody (Synaptic Systems SYSY, Göttingen, Germany, Cat# 218003) diluted in Tris B at 1:200, covered with inverted slide, and all slides were placed in a tightly covered container and incubated overnight at 4 °C.

On the next day, the chambers were opened, and the slides were washed twice in Tris A for 5 min and in Tris B for 10 min. Slides were then placed in incubation chambers filled with the secondary Biotin-SP donkey anti-rabbit IgG specific antibody (Jackson Immuno Research Laboratories, West Grove, PA, USA, #711-065-152) diluted in Tris B/4% donkey normal serum at 1:400. After 1 h at RT, the slides were washed again in Tris A twice for 5 min; then endogenous peroxidase was blocked by incubation in freshly prepared peroxidase blocking solution (10% methanol, 3% H_2_O_2_ in Tris buffer)) for 10 min at RT, and finally washed three times in Tris A for 5 min and once in Tris B for 10 min at RT. 

Vectastain^®^ Elite^®^ ABC kit (Vector Laboratories, Burlingame, CA, USA, cat # PK-6100) was used for enzyme conjugation. ABC reagent mix diluted in Tris B according to the manufacturer’s instructions was prepared 30 min prior to incubation. Slides were chambered and incubated in ABC for 1 h at RT. Chambers were disassembled, and the slides were washed twice for 5 min in Tris, then in dH_2_O for 5 min. Slides were placed face-up on the flat surface to air-dry for a few min. Finally, sections were stained using the DAB (3,3′-diaminobenzidine) substrate kit (Vector Laboratories, Burlingame, CA, USA, # SK-4100). The DAB solution mix was made according to the manufacturer’s instructions, and a few drops were added to each slide to cover the surface. After the color development with similar incubation intervals maintained for each slide, the slides were washed twice in Tris A and once in dH2O for 5 min. Slides were dehydrated in 50%, 70%, 95%, and 100% ethanol for 3 min in each, then cleared with 2 changes of Xylenes (Ibis Scientific, Las vegas, NV, USA, #LC269704) for 10 min each, and coverslipped using the DEPEX (Fisher Scientific, Waltham, MA, USA, #50-980-372) mounting media, both according to manufacturer’s instructions.

For detection of the non-pyroglutamated Aβ plaques, a similar protocol was used with a commercial anti-Aβ monoclonal antibody 6E10 (Biolegend, San Diego, CA, USA, #SIG-39320) at a 1:800 dilution with the secondary Biotin-SP donkey anti-mouse IgG specific antibody (Jackson Immuno Research Laboratories, West Grove, PA, USA, #715-065-151) diluted in TrisB/4% donkey normal serum at 1:400.

### 4.7. Whole-Slide Imaging and Data Analysis

Nine 40 μm sections per brain equally spaced between points +1.18 and −4.24 mm, relative to Bregma, were analyzed for all staining experiments. 

The whole slide imaging for the assays was provided by the Alafi Neuroimaging laboratory at Washington University School of Medicine (St. Louis, MO, USA). The slides were digitized using a whole slide scanner (Nanozoomer 2.0 HT, Hamamatsu, Bridgewater, NJ, USA). The whole slide images were collected through an Olympus UPlanSApo 20×/0.75 lens and captured using the NDP.view2 digital slide viewer software (Hamamatsu Corporation, Bridgewater, NJ, USA). Quantification was performed using a digital pathology software (Visiomorph, Broomfield, CO, USA). 

### 4.8. Confocal Microscopy

Then, 40-μm brain sections from 8 month old 5XFAD mice were stained with rabbit anti-AβpE3 [Pyro Glu3] antibody (SYSY, 1:200) and 6E10 antibody (BioLegend, 1:1000), as described in [46]. Immunofluorescent sections were visualized and captured using an Olympus FX 1200 confocal microscope. Representative images represent confocal Z-stacks (12 slices at 1.79-micron step intervals) taken at 40× magnification, and Z-stacks (12 slices at 1.51-micron step intervals) taken at 20× magnification [46]. 

### 4.9. Data Analysis and Graphing

Statistical analysis was performed by the Mann–Whitney U test using the GraphPad Prism software (version 9, GraphPad Prism, San Diego, CA, USA). 

## 5. Conclusions

The current immunotherapeutic strategies for AD are primarily aimed at the treatment of early symptomatic patients with monoclonal antibodies. A more proactive approach based on immunoprevention of AD in healthy individuals at risk of disease is contingent upon the emergence of novel biomarkers for the prediction of AD prior to pathology onset. Multiple minimally invasive biomarkers are currently being developed, refs. [75,76,77,78,79,80,81] and their application in clinical setting will outline a broad population of otherwise healthy individuals who may benefit prom prophylactic vaccination against AD.

## Figures and Tables

**Figure 1 ijms-24-09797-f001:**
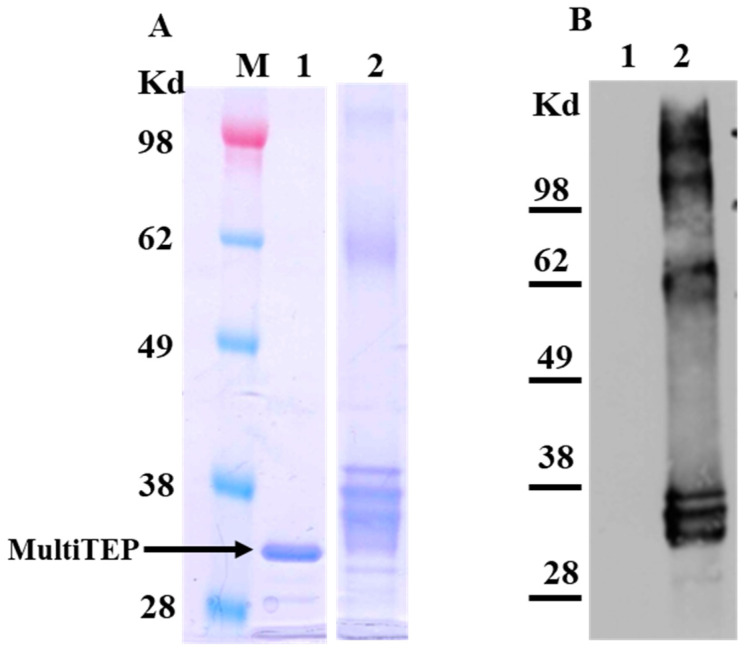
Characterization of the conjugate. Analysis of the carrier and the conjugated product by SDS-PAGE, followed by Coumassie staining (**A**), and western blotting with anti-pE_3_Aβ commercial antibodies (**B**). Note the complete absence of signal in lane 1 of the western blot, indicating that all the bands detected in lane 2 contain the conjugated epitope pE_3_Aβ. M. Marker for molecular weight 1. Unconjugated MultiTEP carrier 2. Conjugation product with pE_3_Aβ.

**Figure 2 ijms-24-09797-f002:**
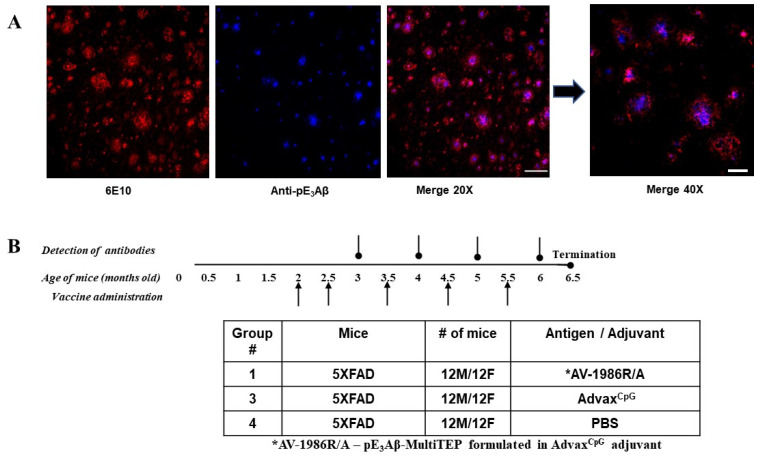
Validation and immunization of transgenic animals. (**A**) Confocal microscopy confirms pyroglutamate-modified pE_3_Ab (blue) peptides within 6E10 positive (red; Aβ_1–17_) Aβ aggregates in a 5XFAD transgenic mouse model of AD. 20× scale bar, 100 µm. Representative 40× scale bar, 50 µm. (**B**) Schematic depicting immunization study of 5XFAD mice injected (arrow) with AV-1986R/A vaccine, Advax^CpG^ adjuvant, or PBS (*n* = 12 M/12F mice per group). * AV-1986R/A is a pE_3_Aβ-MultiTEP formulated in an Advax^CpG^ adjuvant.

**Figure 3 ijms-24-09797-f003:**
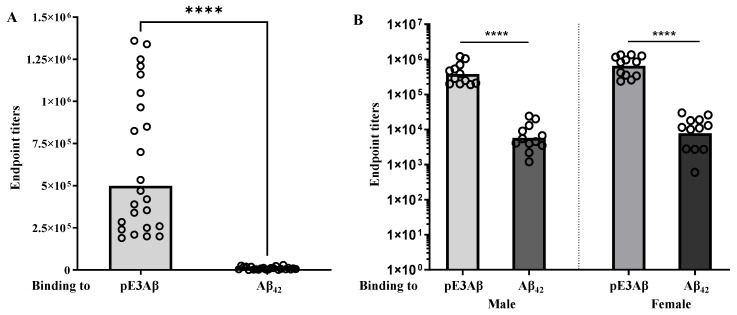
Immunogenicity and selectivity. Endpoint titers of generated antibodies were evaluated in the sera of individual mice (*n* = 24) after the fourth immunization by ELISA. Bars indicate the geometric mean titer, and the *x*-axis indicates the target peptide in ELISA. Titers of generated antibodies specific to full-length Aβ_42_ are significantly lower than titers of antibodies specific to pyroglutamate modified Aβ_3–11_. The ratio of geometric mean titers of anti-pE_3_Aβ to anti-Aβ_42_ was 74:1. (**A**) Splitting of the data by sex showed similar results, with a 66:1 ratio for males (*n* = 12) and 83:1 for females (*n* = 12). (Y-scale is logarithmic) (**B**). 74:1. (**A**) Splitting of the data by sex showed similar results, with a 66:1 ratio for males (*n* = 12) and 83:1 for females (*n* = 12). (Y-scale is logarithmic) (**B**). **** is *p* ≤ 0.0001, symbol “

” presents endpoint titers of Antibodies in the sera of an individual mouse.

**Figure 4 ijms-24-09797-f004:**
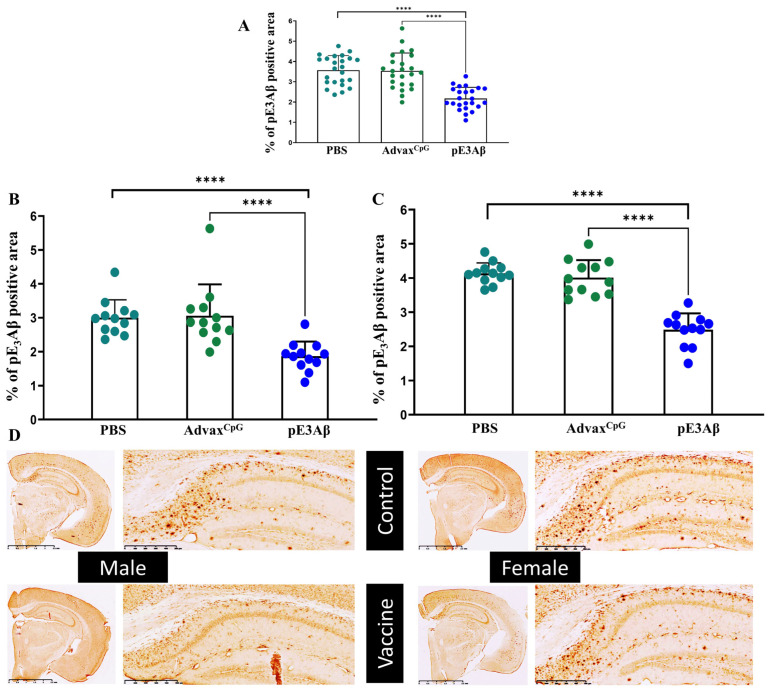
Clearance of pE3-positive plaque from the brains of 5xFAD mice quantified by immunohistochemistry. Brain sections from 5xFAD mice injected with the pE_3_-based vaccine formulated in Advax^CpG^ adjuvant were compared to brain sections from control animals injected with either Advax^CpG^ adjuvant or PBS. The sections were stained with commercial anti-pE_3_Aβ antibodies, and the pyroglutamated plaque-positive area was calculated and plotted. Brain slices from the vaccinated animals contained a significantly less anti-pE_3_Aβ positive area compared to either control with *p* < 0.0001 (****) (**A**). The splitting of data points to males (**B**) and females (**C**) resulted in the same intergroup relationship with *p* < 0.0001(****) for each. No significant differences were observed between the 2 controls. Representative images are shown in panel (**D**), scale bars = 2.5 mm; enlarged hippocampal areas presented, scale bars = 500 µm.

**Figure 5 ijms-24-09797-f005:**
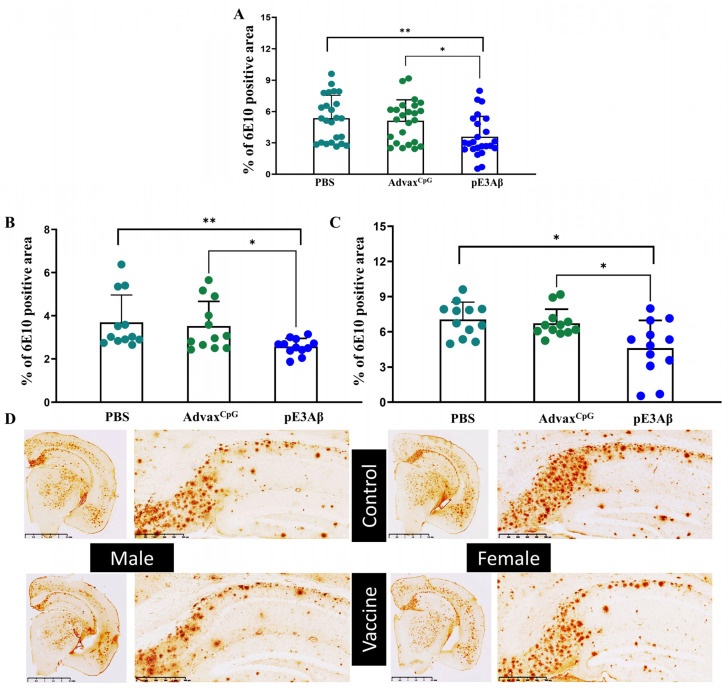
Clearance of non-pyroglutamated Aβ plaque from the brains of 5XFAD mice quantified by immunohistochemistry. Brain sections from 5XFAD mice injected with the pE_3_-based vaccine formulated in Advax^CpG^ adjuvant were compared to the brain sections from control animals injected with either Advax^CpG^ adjuvant or PBS. The sections were stained with commercial anti-Aβ monoclonal antibodies 6E10, and the positive area was calculated and plotted. Brain slices from the vaccinated animals contained significantly less 6E10 positive area compared to either control, with *p* = 0.013 (*) for the Advax^CpG^ control and *p* = 0.003 (**) for the PBS control (**A**). Splitting the data by sex resulted in *p* = 0.033 (*) for Advax^CpG^ in males; *p* = 0.002 (**) for PBS in males (**B**); *p* = 0.016 (*) for Advax^CpG^ in females; *p* = 0.011 (*) for PBS in females (**C**). No significant differences were observed between the two controls. Representative images are shown in panel (**D**), scale bars = 2.5 mm; enlarged hippocampal areas presented scale bars = 500 µm.

**Figure 6 ijms-24-09797-f006:**
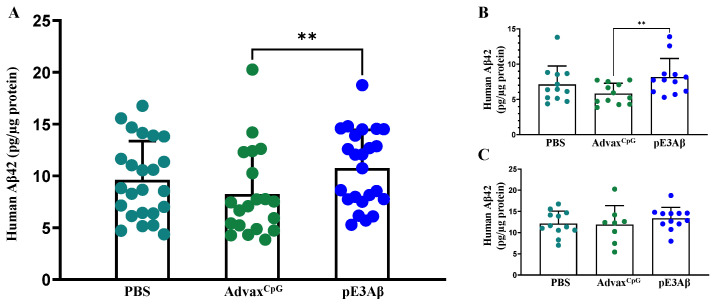
Clearance of soluble Aβ from the brains of 5XFAD mice quantified by ECLIA. Brain tissue samples from 5XFAD mice injected with the pE_3_Aβ-based vaccine formulated in an Advax^CpG^ adjuvant were compared to brain sections from control animals injected with either Advax^CpG^ adjuvant or PBS. The tissue samples were extracted with a T-PER reagent, and the extract was separated by centrifugation into a soluble fraction (supernatant) and an insoluble fraction (pellet). The soluble fraction was normalized by protein content using bicinchoninic acid assay and loaded onto an ECLIA plate. The concentration of human Aβ_42_ was measured and plotted. The vaccine group had a significantly higher level of (** *p* = 0.0076) human Aβ_42_ when compared to the Advax^CpG^ group, but not when compared to the PBS group (**A**). When the animals were split by sex, the effect persisted in males (**B**) with ** *p* = 0.0083, but no significant differences were seen between the groups in females (**C**). Groups injected with the PBS, Advax^CpG^, and pE_3_Aβ-based vaccines are shown in different colors.

**Figure 7 ijms-24-09797-f007:**
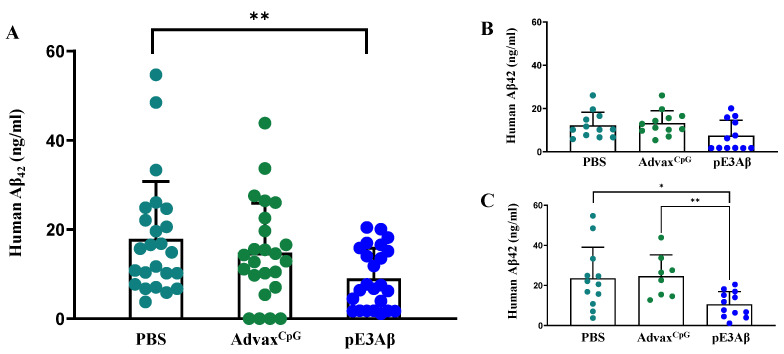
Clearance of insoluble Aβ from the brains of 5XFAD mice quantified by ECLIA. The insoluble pellets obtained from the T-PER extraction were solubilized in formic acid and normalized by the protein concentration in the soluble fraction. The solubilized pellets were then neutralized and loaded onto an ECLIA plate, and the concentration of human Aβ_42_ was measured and plotted. The vaccine group had a significantly lower level of human Aβ_42_ compared to either control, with ** *p* = 0.0047 for PBS and ** *p* = 0.0033 Advax^CpG^ group (**A**). When the animals were split by sex, no significant differences were seen between the groups in males (**B**), while the effect persisted in females, with * *p* = 0.0145 for PBS and ** *p* = 0.0041 for the Advax^CpG^ group (**C**). Groups injected with PBS, Advax^CpG^, and pE_3_Aβ-based vaccines are shown in different colors.

## Data Availability

All data generated or analyzed during this study are included in this article and materials are available from the corresponding author on reasonable request.

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
