# Peer review of "Novel Vaccine against Pathological Pyroglutamate-Modified Amyloid Beta for Prevention of Alzheimer’s Disease"

_ijms, 2023, doi:10.3390/ijms24129797_

Round 1

Reviewer 1 Report

In the manuscript entitled “Novel Vaccine Against Pathological Pyroglutamate-modified 2 Amyloid Beta for Prevention of Alzheimer’s Disease”, authors have prepared a MultiTEP-based vaccine for pE3Aβ and evaluated the efficacy on clearance of soluble and insoluble Aβ from the brain of 5xFAD mice. The background of the study and experimental designs are optimum however the data representation is of poor quality. The following major concern must be addressed before considering the article for proceeding to the next step.

Figure legend-2 is incomplete.

Figure legend-3 is incomplete.

Figure legend-5 is incomplete.

The results from Fig-6 and Fig-7 indicate that, soluble and insoluble fraction of Aβ in the 5XFAD mouse brain determined by ECLIA are gender specific. However, authors mentioned that “The effects of TLR9 agonists are reportedly sex-dependent, which could explain the differences observed in our study [55-57]” which raises two questions?

1.     What is reason behind this gender-specific effect of the vaccine candidate?

2.     How this vaccine candidate will be suitable for all the genders in human?

Plagiarism percentage - not checked by the reviewer.

None

Author Response

Point 1. Figure legend-2 is incomplete. Figure legend-3 is incomplete. Figure legend-5 is incomplete.

Response 1. We thank the reviewer for indicating this omission. We have added more details in the legends of these figures in the revised manuscript.

Point 2. The results from Fig-6 and Fig-7 indicate that soluble and insoluble fractions of Aβ in the 5XFAD mouse brain determined by ECLIA are gender specific. However, the authors mentioned that "The effects of TLR9 agonists are reportedly sex-dependent, which could explain the differences observed in our study [55-57]" which raises two questions?

  1. What is reason behind this gender-specific effect of the vaccine candidate?
  2. How this vaccine candidate will be suitable for all the genders in human?

Response 2. Our findings, depicted in Figures 6 and 7, demonstrate that AD pathology is more prominent in female 5XFAD mice, similar to other transgenic models of Alzheimer's disease (Yang J.T., 2018, Neurosci Bull).

In contrast to the data from this study, in our previous study, we observed a more substantial reduction in insoluble Aβ in the brains of male bigenic 5XFADxPS19 mice compared to females when immunized with the Aβ vaccine formulated with the same adjuvant. The gender-specific differences in vaccine efficacy in mouse models may arise from various factors, including environmental, genetic, and hormonal factors and the severity of amyloid pathology.

Because we did not investigate the underlying reasons for these differences, we believe the made statement about the role of TLR-9 may not have been justified. Consequently, we have removed this statement and included the above paragraph in the revised manuscript in the Discussion section.

Reviewer 2 Report

This is a timely paper, coming as seemingly positive clinical trial results for donanemab have recently been announced. Passive immune therapy for AD now looks feasible, opening the door to vaccines that induce similar immune responses. This paper is a significant contribution to this effort. I have some minor suggestions and questions:

Line 26: 105-106 and 103-104 should be corrected.

50: Words may be missing after "of therapeutically potent".

73-81: You might also note other vaccines that can target pyroGlu-Aβ, such as ACI-24 and TAPAS (https://www.alzforum.org/news/conference-coverage/tapas-anyone-pyroglu-av-vaccine-shrinks-plaques-mice). You could briefly discuss the possible advantages/disadvantages of the various pE3Aβ vaccines, either in the Introduction or Discussion.

119: Change "this" to "these".

Figure 2B: In the table, spell out "Group" instead of abbreviating it "Gr.".

Figure 2: The caption was truncated after "(B) Design of". Please add the rest of the caption.

Figure 2A: I guess "20X" means 20x more than the eyepiece magnification, which might be 10X, for a total magnification of 200X. It seems more meaningful to give the total magnification, including both lenses.

Figure 3: The caption was truncated after "titers of anti-". Please add the rest of the caption.

143: What do you mean by "up to"? Was the mean titer 74-fold higher in one particular subset of samples, but not as much higher in other subsets? Which subsets? For all samples combined, how much higher was the mean titer against pE3Aβ?

150: What do you mean by "¬"?

155: Change "perform" to "performed".

Supplementary Figures: Put them in the same order that you mention them in the text of the paper. Thus put S5 before S3.

Figure 4: It seems to be stretched vertically. 

Figure 4: I assume that the vertical axis is in units of %. You should label it.

Figure 4: I assume that **** means P<0.0001, but you should say so somewhere.

Figure 4: Like in Figure 2A, I guess the magnification stated here doesn't include the eyepiece. It may be more meaningful to the reader to give the total magnification.

175: Eyeballing the graphs suggests that the difference in Figure 5 is either about the same or smaller than in Figure 4. State the numerical values here.

Figure 5: Are the areas in units of %? Label them.

Figure 5: Explain what * and ** mean. I assume P<0.05 and P<0.01, respectively, but you should say so somewhere.

Figure 5: Change "6e10" to "6E10" and "antibodies" to "antibody".

Figure 5: No need for more than 1 or 2 significant digits in P values. More digits don't tell us anything. So use P=0.013 instead of P=0.0126, for example.

Figure 5: The caption is truncated.

Figure 6: It seems to be stretched vertically. 

190: Start a new paragraph to discuss the insoluble fraction.

198: Check the grammar of this sentence.

Figure 7: It seems to be stretched vertically. 

Figure 7: Simplify the vertical axes labels by using units of ng/ml (or ug/L). Then you can drop 3 zeroes from each label.

Figure 7: In the caption, mention A, B, and C in that order--not A, C, and then B.

206-219: Cite references.

207: Since you state the number globally, I don't see a reason to also state the number in one particular country, unless your study focuses on that country. 

213: exist

215: Using the US estimate makes somewhat more sense here than in line 207, but the global estimate would be best.

224: Change the declaration about when donanemab will be approved. You don't know when the article will be published, and people will read your article not only immediately when it is published, but over the next months and years, so take that into consideration when predicting when donanemab will be approved.

225: Change "direIt" to "dire. It".

253: Change "more safe" to "safer".

255: Cite references.

259: Cite references.

261:  Consider changing "immunosenescent" to "in immunosenescent individuals", or something like that.

283: I'm not sure what you meant by "in lieu of" here. Aren't you proposing to use your vaccine for prevention? Maybe say "given the current lack of", or something like that.

292-295: This sentence is confusing. Please re-write.

328: Add "background" before "and".

342: Don't use both "/" and "per". Use only one or the other, as you do on line 344.

349: Move "Fig. 2 (B)." earlier. I suggest putting it after "intramuscularly", in parentheses.

356: State the units for "1000 and 200".

357: What do you mean by "respectively"? The word "respectively" is generally used to connect items between two lists, but the two lists are not evident here.

364: Change ". Fig. S5" to " (. Fig. S5)".

379: Change "/" to " for ".

380: Change "pallets" to "pellets".

382: Change "pallets" to "pellets".

388: Change "pallets" to "pellets".

389: Change "pallet" to "pellet".  Change "Pallets" to "Pellets".

390: Change "/" to " for".

394: Use subscripts in the chemical formulas.

394-395: This sentence is awkward. Re-write.

400: What do you mean by "triplex"? Did you synthesize Aβ triplex? Why? How did you purify it and ensure that it was triplex?

417: Change "ones" to "once".

418: Change "ones" to "once".

420: 230 ml or 230 ul?

422: Change "o" to a degree symbol.

429: Use subscripts in the chemical formulas.

446: 6E10

451: What do you mean by "between Bregma points"? Each mouse has one Bregma point. 

452: What does "approximately 1.18 0.22 and – 4.24 mm" mean?

475-476: I guess that this sentence describes supplementary material for a previous paper. Please update it.

Figure S3 caption: Add "a" before "12-histidine".

The quality of the language is good, though some corrections are needed (see above).

Author Response

We thank the reviewer for thoroughly examining our manuscript and bringing attention to the overlooked mistakes. His/her insightful feedback was extremely valuable in improving the quality of our manuscript. We have included all suggested changes in the revised manuscript.

Q1. Line 26: 105-106 and 103-104 should be corrected.

R.1. The superscripts have been changed as a result of the pdf conversion at the time of submission. We will check the pdf file more carefully.

Q.2 50: Words may be missing after"of therapeutically potent".

R2. A missing word has been added, "of therapeutically potent antibodies."

Q.3. 73-81: You might also note other vaccines that can target pyroGlu-Aβ, such as ACI-24 and TAPAS (https://www.alzforum.org/news/conference-coverage/tapas-anyone-pyroglu-av-vaccine-shrinks-plaques-mice). You could briefly discuss the possible advantages/disadvantages of the various pE3Aβ vaccines, either in the Introduction or Discussion.

R.3. We added information about the optimized version of ACI-24  and "TAPAS" vaccines, and TAP01 humanized antibody in the Introduction section of the revised manuscript.

Q.4. 119: Change "this" to "these."

R.4. Word is Changed

Q.5. Figure 2B: In the table, spell out "Group" instead of abbreviating it "Gr."

R.5. Abbreviation "Gr" was spelled out.

Q.6. Figure 2: The caption was truncated after "(B) Design of". Please add the rest of the caption.

R.6. Truncation of Figure legends occurred due to the pdf conversion at the time of submission. We will check the pdf file more carefully.

Q.7. Figure 2A: I guess "20X" means 20x more than the eyepiece magnification, which might be 10X, for a total magnification of 200X. It seems more meaningful to give the total magnification, including both lenses.

R.7. Thank you for the valuable comment. To clarify the magnification, we added more details in the Methods section of the revised manuscript: "Immunofluorescent sections were visualized and captured using an Olympus FX 1200 confocal microscope. Representative images represent confocal Z-stacks (12 slices at 1.79-micron step intervals) taken at 40× magnification and Z-stacks (12 slices at 1.51-micron step intervals) taken at 20× magnification."

Q.8. 143: What do you mean by "up to"? Was the mean titer 74-fold higher in one particular subset of samples, but not as much higher in other subsets? Which subsets? For all samples combined, how much higher was the mean titer against pE3Aβ?

R.8. We removed the words "up to". The geometric mean titers of antibodies specific to pE3Aβ are 74-fold higher than those specific to Aβ.

Q.9.150: What do you mean by "¬"?

R.9. This is the pdf formatting error and will be corrected in revised version.

Q.10. 155: Change "perform" to "performed."

R.10. "perform" is changed to "performed."

Q.11.Supplementary Figures: Put them in the same order that you mention them in the text of the paper. Thus put S5 before S3.

R.11. The order of supplementary figures was changed according to the order in the text.

Q.12. Figure 4: It seems to be stretched vertically

R.12. All vertically stretched figures were replaced.

Q.13. Figure 4: I assume that the vertical axis is in units of %. You should label it.

R.13. Label of the Y-axis is corrected in all relevant figures.

Q.14. Figure 4: I assume that **** means P<0.0001, but you should say so somewhere.

R.14. Asteriks were defined in all relevant figure legends.

Q.15. Figure 4: Like in Figure 2A, I guess the magnification stated here doesn't include the eyepiece. It may be more meaningful to the reader to give the total magnification.  

R.15. Thank you for the valuable comment, we removed the term “original magnification” from the Fig. 4 and Fig. 5 legends.  The whole slide images were collected through an Olympus UPlanSApo 20x/0.75 lens, displayed on the monitor at optical resolution of 0.46 um/pixel, were captured using NDP.view2 digital slide viewer software and could be zoomed in and out as if you were operating a microscope for the analysis.

For the purpose of clarification we added the following sentence in the Methods section after the information that the slides were digitized using a whole slide software (Nanozoomer  2.0 HT, Hamamatsu, Bridgewater, NJ):  "the whole slide images were collected through an Olympus UPlanSApo 20x/0.75 lens and captured using NDP.view2 digital slide viewer software”.

Since the scale bars are attached to each image and their sizes are specified in the legends, we believe that this could give precise and sufficient information to the reader.

Q.16.175: Eyeballing the graphs suggests that the difference in Figure 5 is either about the same or smaller than in Figure 4. State the numerical values here.

R.16. We agree with the reviewer. We rechecked the numerical values of the graphs and removed the sentence, "Interestingly, these differences had lower statistical significance, yet the mean difference was larger."

Q.17. Figure 5: Change "6e10" to "6E10" and "antibodies" to "antibody."

R.17. "6e10" was changed to "6E10" throughout the manuscript.

Q.18. Figure 5: No need for more than 1 or 2 significant digits in P values. More digits don't tell us anything. So use P=0.013 instead of P=0.0126, for example.

R.18. P values were corrected.

Q.19. 190: Start a new paragraph to discuss the insoluble fraction.

R.19. New paragraph was started for discussion of the insoluble fraction.

Q.20. 198: Check the grammar of this sentence.

R.20. The sentence was changed as follows: "This is, in general, a limitation that can be seen with various mouse models of AD that are only partially replicating very complex human Alzheimer's disease."

Q.21. Figure 7: Simplify the vertical axes labels by using units of ng/ml (or ug/L). Then you can drop 3 zeroes from each label.

R.21. Y-axis labels were changed to ng/ml.

Q.22. Figure 7: In the caption, mention A, B, and C in that order--not A, C, and then B.

R.22. The order was changed to A,B and C in the caption of Fig.7.

Q.23. 206-219: Cite references.

R.23. References are cited in all recommended parts of the revised manuscript.

Q.24. 207: Since you state the number globally, I don't see a reason to also state the number in one particular country, unless your study focuses on that country. 

R.24. As recommended, we stated the number globally here and made a US statement in line 215.

Q.25. 213: exist

R.25. The word is corrected.

Q.26. 224: Change the declaration about when donanemab will be approved. You don't know when the article will be published, and people will read your article not only immediately when it is published, but over the next months and years, so take that into consideration when predicting when donanemab will be approved.

R.26. We agree with the reviewer and removed the timing.

Q.27. 225: Change "direIt" to "dire. It".

R.27. Space was added.

Q.28. 253: Change "more safe" to "safer."

R.28. "more safe" is changed to "safer."

Q.29. 261:  Consider changing "immunosenescent" to "in immunosenescent individuals", or something like that.

R.29. "immunosenescent" is changed to "in immunosenescent individuals."

Q.30. 283: I'm not sure what you meant by "in lieu of" here. Aren't you proposing to use your vaccine for prevention? Maybe say "given the current lack of", or something like that.

R.30. As recommended by the reviewer, we changed the sentence as follows: This is especially promising considering the current lack of preventive vaccinations for people at risk of AD.

Q.31. 292-295: This sentence is confusing. Please re-write.

R.31. We made the paragraph clear by modifying it as follows: Prior reports on the structures of oligomeric and fibrillar species of amyloid showed that the region between residues 1 and 14 are highly surface-exposed and the tyrosine 10 is solvent exposed in Aβ42 oligomers to a similar extent to that found in the unfolded monomer. These data and the epitope prediction tool described in Larsen et al. were used to determine and select the target peptide length.

Q.32. 328: Add "background" before "and".

R.32. We added "background".

Q.33. 342: Don't use both "/" and "per". Use only one or the other, as you do on line 344.

R.33. We removed "per."

Q.34. 349: Move "Fig. 2 (B)." earlier. I suggest putting it after "intramuscularly", in parentheses.

R.34. Fig.2 B was moved.

Q.35. 356: State the units for "1000 and 200".

R.35. Dilutions were presented as "1:1000" and "1:200."

Q.36. 357: What do you mean by "respectively"? The word "respectively" is generally used to connect items between two lists, but the two lists are not evident here.

R.36. The sentence was corrected as follows: Sera samples were diluted 5-fold serially starting at 1:1000 and 1:200 for detection of anti-pE3Aβ and anti-Aβ42, respectively (in 0.3% dry, non-fat milk in TBST, 100 μl/well), then added to the plates and incubated o/n at 4°C.

Q.37. 364: Change ". Fig. S5" to " (. Fig. S5)".

R.37. Changed.

Q.38. 379: Change "/" to " for. "

R.38. Changed throughout the manuscript.

Q.39. 380: Change "pallets" to "pellets."

R.39. This typo was changed throughout the manuscript.

Q.40.  394: Use subscripts in the chemical formulas.

Q.40. We made subscripts in all chemical formulas.

Q.41.  394-395: This sentence is awkward. Re-write.

R.41. The sentence was changed: "Prior to analyses, neutralization buffer (1 M Tris base, 0.5 M NaH2PO4, 0.05% NaN3) was added to the normalized insoluble fractions. The pH of the samples was adjusted to 7 using 5N NaOH. 10,000-fold dilutions of the neutralized samples were used for biochemical analyses." 

Q.42. 400: What do you mean by "triplex"? Did you synthesize Aβ triplex? Why? How did you purify it and ensure that it was triplex?

R.42. We apologize for the confusion. Writing "Triplex," we meant that three species of Aβ peptide (Aβ38, Aβ40, and Aβ42) could be analyzed simultaneously with this kit. We replaced "Triplex" with the commercial name of the kit, "V-PLEX human Aβ peptide panel."

Q.43. 417: Change "ones" to "once".

R.43. Typo was corrected.

Q.44. 420: 230 ml or 230 ul?

R.44. Typo was corrected.

Q.45. 422: Change "o" to a degree symbol.

R.45. Typo was corrected.

Q.46. 451: What do you mean by "between Bregma points"? Each mouse has one Bregma point. 

R.46. We apologize for the confusion. We meant two points relative to Bregma. The sentence was changed to clarify: Nine 40 μm sections per brain equally spaced between points +1.18 and -4.24 mm relative to Bregma were analyzed for all staining experiments.

Q.47. 475-476: I guess that this sentence describes supplementary material for a previous paper. Please update it.

R.47. Sentence was removed.

Q.48. Figure S3 caption: Add "a" before "12-histidine".

R.48. "a" is added before "12-histidine".

Round 2

Reviewer 1 Report

Authors have addressed all the points raised by the reviewer. Therefore, this manuscript should be accepted for publication.

Plagiarism percentage - not checked by the reviewer. 

Author Response

The manuscript was checked by a colleague fluent in English writing to improve the language.